# A Proof of Principle for the Detection of Viable *Brucella* spp. in Raw Milk by qPCR Targeting Bacteriophages

**DOI:** 10.3390/microorganisms8091326

**Published:** 2020-08-31

**Authors:** Michaela Projahn, Jens A. Hammerl, Ralf Dieckmann, Sascha Al Dahouk

**Affiliations:** Department of Biological Safety; German Federal Institute for Risk Assessment, 10589 Berlin, Germany; jens-andre.hammerl@bfr.bund.de (J.A.H.); ralf.dieckmann@bfr.bund.de (R.D.); sascha.al-dahouk@bfr.bund.de (S.A.D.)

**Keywords:** Brucella, bacteriophages, viruses, indicator organisms, helicase, PCR, dairy products, food safety

## Abstract

Brucellosis is still a global health issue, and surveillance and control of this zoonotic disease in livestock remains a challenge. Human outbreaks are mainly linked to the consumption of unpasteurized dairy products. The detection of human pathogenic *Brucella* species in food of animal origin is time-consuming and laborious. Bacteriophages are broadly applied to the typing of *Brucella* isolates from pure culture. Since phages intracellularly replicate to very high numbers, they can also be used as specific indicator organisms of their host bacteria. We developed a novel real-time PCR (qPCR) assay targeting the highly conserved helicase sequence harbored in all currently known *Brucella*-specific lytic phages. Quality and performance tests determined a limit of detection of <1 genomic copy/µL. In raw milk artificially contaminated with *Brucella microti*, Iz_v_ phages were reliably detected after 39 h of incubation, indicating the presence of viable bacteria. The qPCR assay showed high stability in the milk matrix and significantly shortened the time to diagnosis when compared to traditional culture-based techniques. Hence, our molecular assay is a reliable and sensitive method to analyze phage titers, may help to reduce the hands-on time needed for the screening of potentially contaminated food, and reveals infection risks without bacterial isolation.

## 1. Introduction

Brucellosis is a zoonotic disease usually transmitted from animals to humans by direct contact with infected livestock or through the consumption of derived products such as raw meat, milk, and cheese [1,2,3,4,5]. Various species of the genus Brucella have been described but human infections are mainly caused by *B. melitensis, B. abortus* and *B. suis* [6,7,8]. Brucellae can be isolated from products of animal origin using a selective culture medium such as Farrell’s medium. Classical microbiological methods (oxidase, catalase, agglutination with monospecific sera, CO_2_ requirement, H_2_S production, urease activity, growth inhibition by thionin and basic fuchsin, phage-typing) and PCR assays are subsequently used for identification and sub-differentiation of species [9,10]. However, isolation from dairy products using culture methods is time-consuming (taking at least three days up to three weeks) and laborious due to the endogenous milk flora which may overgrow the pathogens. Serological tests like the milk ring test suffer from low specificity due to cross-reactivity with various other bacteria [10,11]. PCR assays are faster than culture and are more specific and sensitive than serological tests [12,13]. Due to DNA amplification, the detection limit can be as low as a single copy of the respective target sequence. The quality of PCR results, however, depends on DNA isolation methods and PCR inhibitors in the matrix, which may reduce sensitivity and can lead to false-negative results [5,13,14]. Furthermore, these methods cannot discriminate between live and dead bacteria which may hamper the evaluation of a positive test result in case of DNA persistence after successful therapy or through contaminations [15,16]. Therefore, a combination of serological and PCR-based methods is highly recommended for epidemiological investigations [13,17,18].

In diagnostic laboratories, bacteriophages are routinely used for subtyping of *Brucella* species and biovars because they show host specificity despite high genomic similarity within the genus [19,20,21,22,23,24]. However, phage concentrations cannot be easily determined due to inconsistent lysis efficiency of the various *Brucella* phages and their differing ability to produce clear plaques. In our study, we therefore developed a qPCR assay for the reliable detection and quantification of all known lytic *Brucella* phages. 

In the last decades, phages have been increasingly considered as tools for detection and control of pathogenic bacteria in food production [25,26,27,28,29,30]. The application of phages is of advantage because they specifically prove the presence of viable bacteria needed for their own replication. Hence, we used *Brucella* phages in combination with the newly established qPCR assay to detect viable brucellae in raw milk.

## 2. Materials and Methods 

### 2.1. Phage Propagation

*Brucella* phages Iz_v_, Bk2_v_ and Wb_v_ were propagated on *Brucella* (*B*.) *microti* 4910^T^ using available phage supernatant stocks as previously described [24]. *Brucella microti* was chosen to allow for test application under biosafety level 2 conditions. In brief, *B. microti* was grown in *Brucella*-broth to McFarland 1–2 in ventilated T250 cell culture flasks (VWR International GmbH, Darmstadt, Germany). Phage stocks were added to a multiplicity of infection (MOI) of 1 to 0.01 and incubated aerobically at 37 °C on a rotational shaker for 24–48 h. Lysates were centrifuged for 30 min at 8,500× *g* to remove bacterial debris and the supernatant was sterile filtered (0.2 µm). Afterwards, samples were treated with DNaseI/RNaseA (20 µg/mL *wt*/*vol*; Roche, Mannheim, Germany) for 30 min at 37 °C. Phage titers were determined by spot tests on *B. microti* soft-overlay agar (10 µL phage lysate of 10-fold dilutions) and plaque tests (100 µL phage lysate of 10-fold dilutions) as previously described [24]. All assays were run in duplicates of two independent experiments. 

### 2.2. Phage DNA Isolation and Quantification

For DNA isolation, Iz_v_ phage was enriched using polyethylene glycol (PEG) precipitation. PEG6000 (Fluka Biochemica, St. Gallen, Switzerland) was added to the phage lysate to a final concentration of 10% and NaCl to a final concentration of 1 M. The suspension was incubated for 3 h at 4 °C and then centrifuged for 20 min at 8,500× *g* and 4 °C. DNA was isolated from the resulting pellet with the PureLink™ Genomic DNA Mini Kit (Invitrogen, Fisher Scientific GmbH, Schwerte, Germany) according to manufacturer’s instructions. DNA concentration was measured using the Invitrogen Qubit™ fluorometer (Fisher Scientific GmbH, Schwerte, Germany). Genomic copy numbers were calculated with an open access online tool (http://cels.uri.edu/gsc/cndna.html, Genomics and Sequencing Center, University of Rhode Island) assuming a template length of 41,446 bp for the Iz_v_ phage (GenBank accession no. KY056619.1). The phage DNA was serially diluted in PureLink™ Elution Buffer (Invitrogen) with final concentrations of 1 × 10^6^ to 1 × 10^0^ genomic copies/PCR reaction.

### 2.3. Brucella Phage Helicase (BPHeli) qPCR Assay Design and Running Conditions

Primers were designed with Primer3 (http://bioinfo.ut.ee/primer3/) [31,32] using the helicase as target (accession no. KY056619.1, Figure 1) because this gene showed 100% sequence identity among all available genomes of lytic *Brucella* phages [24,33,34] and was assumed to be present in all other known non-reference phages.

Primers were analyzed for standard parameters like melting temperature (Tm) and guanine-cytosine (GC)-content as well as probability of dimer creation using the Multiple Primer Analyzer (Thermo Fisher Scientific, Waltham, MA, USA). Furthermore, specificity and coverage were checked against the whole NCBI database using BLAST (https://blast.ncbi.nlm.nih.gov/Blast.cgi). Our primers turned out to be 100% specific, with 100% coverage exclusively for the lytic *Brucella* phages. No unspecific binding sites in other bacterial or viral genomes were found. The total reaction mixture of 25 µL contained 12.5 µL of QuantiFast SYBR Green Mastermix (Qiagen, Hilden, Germany), 1 µL of each primer (BPHeli(+1226): 5′-ctggacactatggaaggcgg-3′, BPHeli(+1399): 5′-gccgttcattggggtctaca-3′, 10 µM), 2 µL of the sample and 8.5 µL PCR water. The qPCR was carried out with a CFX96 Touch Real-time PCR machine (BioRad, Hercules, CA, USA) choosing the following settings: 5 min of initial denaturation at 95 °C, 40 cycles of 10 s at 95 °C and 30 s at 60 °C (fluorescent measurement at the end of each cycle), with a final melting curve analysis. 

### 2.4. Performance Indicators of the BPHeli qPCR Assay 

Analytical sensitivity of the BPHeli qPCR assay was determined using Iz_v_ phage DNA purified and serially diluted as described above. Quantification cycle (Cq) values were plotted against the respective number of genome equivalents, and standard deviations and log-linear regression were calculated using IBM SPSS Statistics v. 21. In addition, phage lysates were analyzed to test the applicability of the assay to quantify phage concentrations in propagation experiments. Standards were included in each run. Specificity was tested both in silico using BLAST against the NCBI database and in vitro using lysates of the 19 *Brucella* phages available, namely Iz_v_, Wb_v_, Bk2_v_, F25, Tb_v_, R/C_v_, F25u, F1m, P, 371/XXIX, F1u, 12m, 3, MP51, 6, 45/III, L19, Fi_v_, and F1 [24]. Genomic DNA of *B. microti* 4910^T^, *B. abortus* 544, and *B. melitensis* 16M were used as negative controls to exclude cross-reactivity and inhibitory effects in the analysis of phage lysates. 

Performance of the BPHeli assay was also tested in raw cow’s milk to exclude inhibitory effects of the food matrix. To this end, we prepared log10 dilutions of Iz_v_ phage DNA in raw cow’s milk (1:10, 1:5, and 1:2 in DNA elution buffer). Furthermore, we assessed the ability of the BPHeli qPCR to directly detect Iz_v_, Bk2_v_, and Wb_v_ phage lysates in milk by comparing the molecularly detected concentrations in PFU/mL with the concentrations determined by spot and plaque tests.

### 2.5. Stability of the BPHeli qPCR Assay in Cow’s Milk

Iz_v_ phage lysates were diluted in 50 mL *Brucella*-broth (BrucB), selective *Brucella*-broth (BrucSB), raw cow’s milk 1:10 in *Brucella*-broth (MBrucB) and raw cow’s milk 1:10 in selective *Brucella*-broth (MBrucSB), to a final concentration of 10^5^ PFU/mL. Higher concentrations of the Iz_v_ phage were used to facilitate the monitoring of changes during the experiments. The mixtures were incubated aerobically in ventilated T250 cell culture flasks at 37 °C for 39 h and subsamples for qPCR were taken after 0, 13, 15.5, 17.5, 22, and 39 h.

### 2.6. Phage Propagation and Growth of B. Microti in Raw Cow’s Milk

Iz_v_ phage lysates were diluted in each of the four broth mixtures as described above. *Brucella microti* 4910^T^ was added to final concentrations of 10^1^ and 10^2^ cfu/mL. A *B. microti* control was prepared for all mixtures (10^2^ cfu/mL). The flasks were incubated at 37 °C for 39 h and subsamples for qPCR were taken after 0, 13, 15.5, 17.5, 22, and 39 h. Changes in phage concentrations were measured using our newly developed BPHeli qPCR assay, while *B. microti* concentrations were determined by the genus-specific IS711 qPCR, as published by Matero and colleagues with slight modifications [35,36]. Briefly, we used the QuantiFast Pathogen Kit (Qiagen, Hilden, Germany) according to manufacturer’s instructions. The reaction mixture contained 0.5 µLprobe, 1 µL of each primer using 10 µM working solutions and 5 µl DNA template in a final volume of 25 µL. The PCR was carried out in a CFX96 Touch Real-Time PCR Detection System (BioRad) at 95 °C for 5 min, 45 cycles of 15 s at 95 °C and 30 s at 60 °C, and a final step at 60 °C for 1 min.

## 3. Results

Our study pursued two major objectives: (i) to develop a fast and reliable assay for the detection of *Brucella* phages and for monitoring phage titers in propagation experiments, and (ii) to apply phage detection as an indirect evidence of the presence of viable bacterial pathogens in food using *Brucella* in raw cow’s milk as a model. 

### 3.1. Phage Propagation and DNA Isolation

We selected three *Brucella* phages with a broad host range for our experiments. After propagation of Iz_v_, Bk2_v_ and Wb_v_ on *B. microti* reference strain 4910^T^, titers of 4 × 10^7^, 2.8 × 10^6^ and 6 × 10^6^ PFU/mL were determined in the spot assay, and titers of 2.1 × 10^7^, 7.1 × 10^5^, and 1.8 × 10^6^ PFU/mL in the plaque assay, respectively. *Brucella microti* was used to facilitate lab work under biosafety level 2. Since *Brucella* phages are highly conserved, we only isolated DNA from the purified Iz_v_ phage (resulting in 30 ng/µL).

### 3.2. Brucella Phage Helicase (BPHeli) qPCR Assay Performance

Serial 1:10 DNA dilutions from 10^6^ to 10^0^ Iz_v_ phage genome copies were used to assess the analytical sensitivity of our newly developed BPHeli qPCR assay. Data were obtained from 17 independent runs. Cq (cycle of quantification)-values were plotted against the respective numbers of genome equivalents, and standard deviations and log-linear regression were calculated as performance indicators using IBM SPSS Statistics v. 21 (IBM Corp., Armonk, NY, USA) (Figure 2, Table 1). 

The limit of detection was determined to be lower than 1 genomic copy/µL. There was a log-linear correlation between Cq values and the number of phage genome equivalents. The standard deviations of the mean Cq values ranged between 0.17 and 0.19 with slightly higher values at lower bacteriophage concentrations (<10^2^ Iz_v_ phage genome equivalents). The detection of 10^0^ genome equivalents failed in three runs. Positive fluorescence signals were routinely checked by melting curve analysis. 

For specificity testing, lysates of 19 different *Brucella* phages as well as *Brucella* spp. genomic DNA were tested. Using the BPHeli qPCR all *Brucella* phages could be detected. Although genomic sequences were only available for a few *Brucella* phages, our findings showed that the chosen target region is highly conserved among the phages. Calculation of the phage DNA concentration in the lysates of Iz_v_, Bk2_v_ and Wb_v_ revealed 6 × 10^9^, 1.1 × 10^10^, and 1.2 × 10^8^ PFU/mL, respectively. The control samples were tested negative in the qPCR assay. Unspecific amplification signals only occurred in phage-free samples and could be easily identified by melting curve analysis (expected melting point at 85.5–86 °C, false-positive samples at 76.5 °C; Figure 3). Primer-dimer formation (self-dimers, cross-dimers) was excluded in silico as well as by melting curve analysis of negative control samples.

To assess a possible matrix effect on the performance of the BPHeli qPCR assay, Iz_v_ phage DNA was diluted in raw cow’s milk. Phage DNA was detected in all milk samples independent of the concentrations introduced. However, there was a fourfold increase of Cq-values as well as a reduction in the relative fluorescence units (RFU) of the amplification curves with an increasing proportion of milk in the test sample. 

### 3.3. BPHeli qPCR Assay Stability in Cow’s Milk

The Iz_v_ phage lysate was added to four different broth mixtures to a final concentration of 10^5^ PFU/mL based on previous qPCR results. Results from the BPHeli qPCR assay indicated reduced amounts of phage target genomes in the broth (up to 3 log10 values) which was most likely a matrix effect because samples from the mixtures were directly used for PCR without preceding DNA extraction (Figure 4). 

The matrix effect could not be exclusively explained by the added raw milk because there was also a decrease in the phage concentration (approx. 1 log10) when BrucB (*Brucella* broth) and BrucSB (selective *Brucella* broth) were compared. The measured Iz_v_ phage concentration remained stable over 39 h except for MBrucB (raw cow’s milk in *Brucella*-broth) (Figure 4). After 13 h, we observed a decline of the phage concentration in MBrucB of about 1 log10 value.

### 3.4. Growth of B. microti and Iz_v_ Phage Propagation in Raw Cow’s Milk

Multiplication of *B. microti* 4910^T^ and Iz_v_ phage in raw cow’s milk was determined using qPCR assays for the detection of *Brucella* and phage DNA, respectively. The experiments were carried out in two different *Brucella* media with and without adding raw cow’s milk (1:10 dilution in the respective medium). Iz_v_ was used because it has the widest lysis spectrum among *Brucella* spp. and, therefore, the greatest application potential.

*Brucella microti* growth and propagation of Iz_v_ phage were measured at six points in time over a period of 39 h (Figure 5). 

Growth of *B. microti* (C + D + E) could be detected in BrucB (I) as well as in BrucSB (III) after 13 h. However, growth in BrucSB (III) was slower and the final bacterial concentration after 39 h of incubation was approx. 2–3 log10 values lower. Furthermore, samples with starting concentrations of 10^2^ cfu/mL (D + E) revealed a larger increase in *B. microti* concentrations compared to the 10^1^ cfu/mL samples (C). In the BrucB sample (I) with 10^2^ cfu/mL (free of phages), the growth of *B. microti* (E) seemed to slow down after 17.5 h of incubation which could not be confirmed in the BrucSB sample (III). Iz_v_ phage propagation in BrucB (I) was detected after 17.5 h in the 10^2^ cfu/mL sample (B) and after 22 h in the 10^1^ cfu/mL sample (A). After these phage-bursts there was still bacterial growth in the samples as indicated by increasing cfu/mL (D + E). The propagation of the Iz_v_ phage in BrucSB (III) could not be detected before 22 h (10^2^ cfu/mL, B) and 39 h (10^1^ cfu/mL, A) of incubation. Final Iz_v_ phage DNA concentrations differed approx. 2 log10 values between BrucB (I) and BrucSB (III). 

*Brucella microti* growth in raw cow’s milk (1:10 diluted in *Brucella* broth, MBrucB, II) could have been observed only during the first 13 h of the incubation period in two out of three samples with higher bacterial starting concentrations (D + E) whereas in all the raw milk samples diluted in selective *Brucella*-broth (1:10, MBrucSB, IV) an additional slight increase of the bacterial concentration after 22 h was determined (C + D + E). Propagation of the Iz_v_ phage was not detected in MBrucB samples (II, A + B). The phage concentration seemed to decrease right at the beginning of the incubation period and remained more or less stable after 13 h of incubation. In contrast, in the MBrucSB sample with 10^2^ cfu/mL (IV, B) a sharp rise of about 6 log10 values was observed after 39 h of incubation. A comparison of qPCR results with plaque or spot tests was impossible due to overgrowth by the endogenous milk flora.

## 4. Discussion

We developed a real-time PCR assay for the detection of 19 different *Brucella* phages. A previously published PCR targeting the DNA polymerase gene of phage S708 was only tested for four out of them, namely Tb, Bk, Fz, and Wb [37]. Although our newly established BPHeli qPCR performed well, the phage concentrations (PFU/mL) determined by spot test, plaque test and the molecular assay varied, also depending on the phages under study. This observation could be partially explained by the amplification of phage DNA originating from non-functional or damaged phage particles using the qPCR assay [38]. Furthermore, some of the *Brucella* phages did not produce clear plaques, which makes the assessment of exact phage titers of a lysate technically demanding. Phage particles may also stick together resulting in different sizes of spots and plaques leading to higher variations in PFU counts [38,39]. In contrast, our qPCR assay revealed less variability than the biological tests and provided results within two hours, whereas spot and plaque tests took up to two days of incubation. 

To evaluate our newly developed phage-based qPCR assay for the detection of bacteria in complex food matrices, we initially conducted stability and performance tests and then analyzed raw cow’s milk contaminated with *Brucella*. Our study showed that qPCR results, and thus indirectly phages, remained stable over 39 h of incubation, which makes *Brucella* phages a useful tool for the indirect detection of viable brucellae. The BPHeli qPCR assay was able to detect *Brucella* phages in raw milk samples without any preceding DNA isolation. By omitting the DNA pre-processing step, we could significantly reduce hands-on-time and speed up time-to-result. However, assay sensitivity was lower in raw cow’s milk than in culture medium, which might have been a result of PCR inhibitors in the food matrix, but DNA isolation does not necessarily improve the analytical sensitivity of qPCR assays [14]. 

Indirect detection of bacteria by phages is based on the addition of phages or phage cocktails to a target sample and subsequent analysis of phage propagation [40,41]. In our study, we inoculated raw cow’s milk with *B. microti*, added *Brucella* phages and used a qPCR assay to evaluate the moment of the phage burst in the sample. Thereby we tested whether *Brucella* phages can serve as a proof of viable *Brucella* cells in raw milk. Enrichment culture and the isolation of *Brucella* spp. from raw milk products is laborious and time consuming taking up to several weeks. Because of the natural milk microbiome selective media are needed and periodic sub-culturing has to be performed [9]. Usually serological and molecular methods are combined to prove the presence of *Brucella* in raw milk products [17]. The application of phages can accelerate diagnosis compared to traditional culture methods used for bacterial isolation from raw milk products. As phages only propagate in viable bacterial cells, their detection also provides more meaningful results on the contamination of dairy products with *Brucella* spp. Using IS711 and BPHeli qPCR assays simultaneously allowed for tracking the growth of *B. microti* and the propagation of the Iz_v_ phage in raw cow’s milk, respectively. As anticipated, *B. microti* was able to grow slowly in a milk-broth mixture including selective medium (MBrucSB) whereas bacteria did not grow using non-selective medium (MBrucB). Under the latter conditions, the natural milk microbiome inhibited or prevented successful bacterial growth. After 22 h of incubation, the number of *B. microti* cells in MBrucSB measured by IS711 qPCR has risen by 1 to 2 log10 values. The detection of IS711 is very sensitive since this genetic element occurs in multiple copies in the *Brucella* genome. However, the number of IS711 elements in *B. microti* 4910^T^ is higher (*n* = 13, Genbank CP001578.1 + CP001579.1) than in *B. melitensis* 16M (*n* = 7, Genbank CP007763.1 + CP007762.1) or *B. abortus* S19 (*n* = 4, Genbank CP000887.1 CP000888.1). This might lead to an earlier detection of the growth of *B. microti* compared to other, *Brucella* spp. Hence, the reliability of the IS711 qPCR assay for an early detection of *Brucella* spp. multiplication in raw milk must be questioned. In contrast, at 39 h of incubation the reproduction of Iz_v_ phages was more clearly visible with concentrations changing from 10^2^ to 10^8^ PFU/mL. In a previous study, this phage burst appeared after 48–72 h depending on the *B. abortus* concentration in the sample [37]. The observed discrepancy might be explained by diverging sampling schedules (leading to a later detection) or by the use of different *Brucella* spp. in the experiments. Classical *Brucella* spp. are slow-growing bacteria while *B. microti* is known to grow fast, within 1–2 days on standard media [42,43]. 

Sergueev and colleagues used the S708 phage for their diagnostic experiments in clinical samples whereas we decided to use the Iz_v_ phage because its host range comprises all known *Brucella* spp. relevant for human infections [24]. S708 is lytic for *B. suis*, *B. abortus*, and *B. neotomae* [34], but lacks activity against *B. melitensis*. However, *B. melitensis* is the most common cause of human infections worldwide and is highly related to outbreaks due to the consumption of contaminated dairy products from goats and sheep [44,45]. The detection of *Brucella* in raw milk products using phages should, therefore, comprise all major zoonotic *Brucella* spp.

In our experiments, we further observed that even after the phage burst bacterial cells were still viable and culturable implying that Brucella might not be fully eradicated from a food sample using phages or gained phage resistance [46]. To overcome resistance, phage cocktails can be applied [47]. Moreover, phage propagation might depend on a certain number of viable bacterial cells as the phage burst only occurred after 17.5 h and 22 h of incubation in BrucB and BrucSB, respectively. As already shown, the starting concentration of *Brucella* phages also influences the timing of the phage burst [37]. *Brucella* phages display a varying host range, even though they are genetically closely related [24]. Except for the genomic information, there is still a huge lack of knowledge on gene translation and regulation, protein structures, phage-host interactions, and the general propagation cycle of these phages. For the *Escherichia coli* T7 phage more comprehensive information is already available, making it a useful tool for pathogen detection and phage display experiments [48,49]. Hence, there is an urgent need for further molecular and biological investigations to better understand *Brucella* phages and to evaluate their application in the context of food safety. 

## 5. Conclusions

We developed a highly sensitive and robust qPCR assay for the detection of *Brucella* phages both in lysates and in raw milk and products thereof. In this way, our assay indirectly proves the presence of viable *Brucella* spp. after adding *Brucella* phages to a contaminated food sample, which is a much faster approach than traditional culture methods. Rapid molecular diagnostics, which are capable of detecting live bacteria, allow for better risk assessment and may significantly improve food safety standards in the future.

## Figures and Tables

**Figure 1 microorganisms-08-01326-f001:**
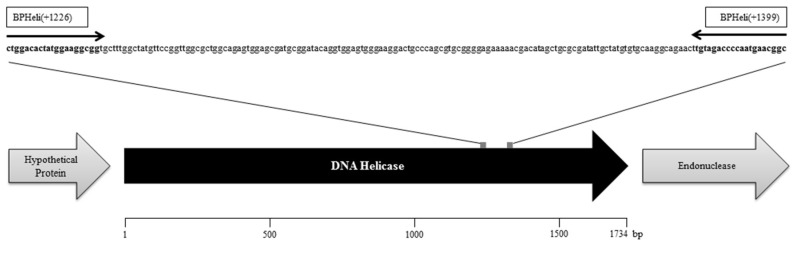
Consensus sequence (174 bp) and primer binding sites of the Brucella Phage Helicase (BPHeli) qPCR assay using accession no KY056619.1 as reference.

**Figure 2 microorganisms-08-01326-f002:**
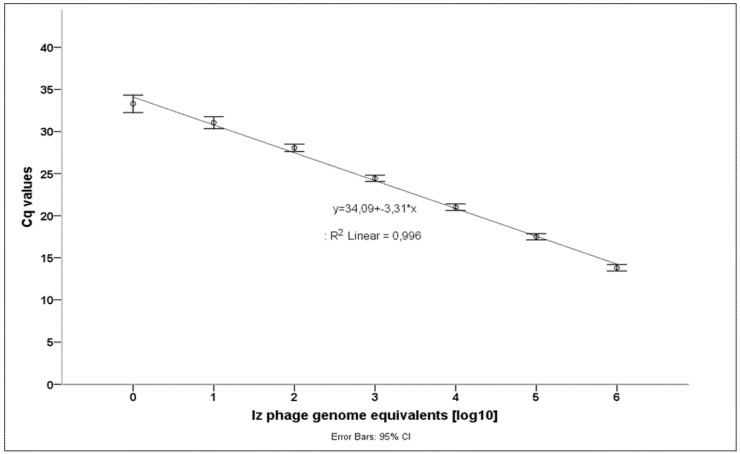
Sensitivity testing using 10fold serial dilutions of genomic Iz_v_ phage DNA. A calibration curve was calculated from 17 independent PCR runs. Cq = cycle of quantification.

**Figure 3 microorganisms-08-01326-f003:**
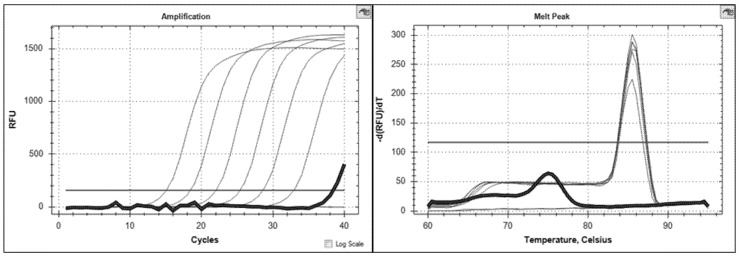
BPHeli qPCR amplification curves and melting peaks of a standard curve (melting point at 85.5–86°C) with one true-negative control (no melting peak) and one false-positive negative control highlighted in bold (melting point at 76.5 °C).

**Figure 4 microorganisms-08-01326-f004:**
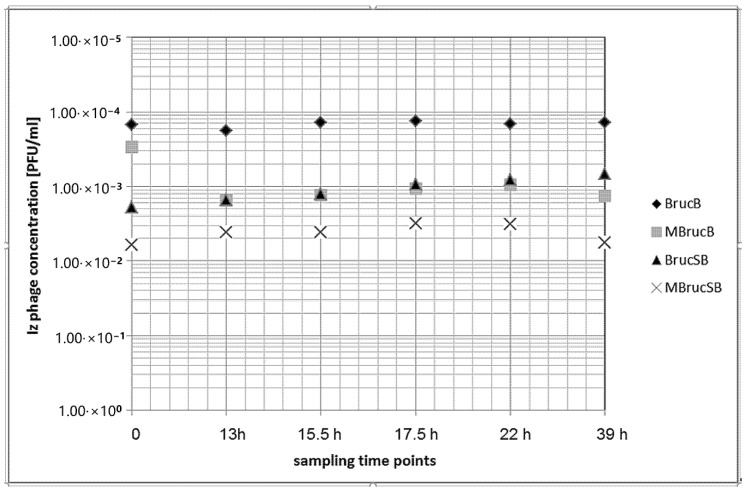
Stability of Iz_v_ phage in four different broth mixtures measured by the BPHeli qPCR assay over a time period of 39 h at 37 °C (BrucB—Brucella-broth, MBrucB—raw cow’s milk in Brucella-broth (1:10), BrucSB—selective Brucella-broth, MBrucSB—raw cow’s milk in selective Brucella-broth (1:10)).

**Figure 5 microorganisms-08-01326-f005:**
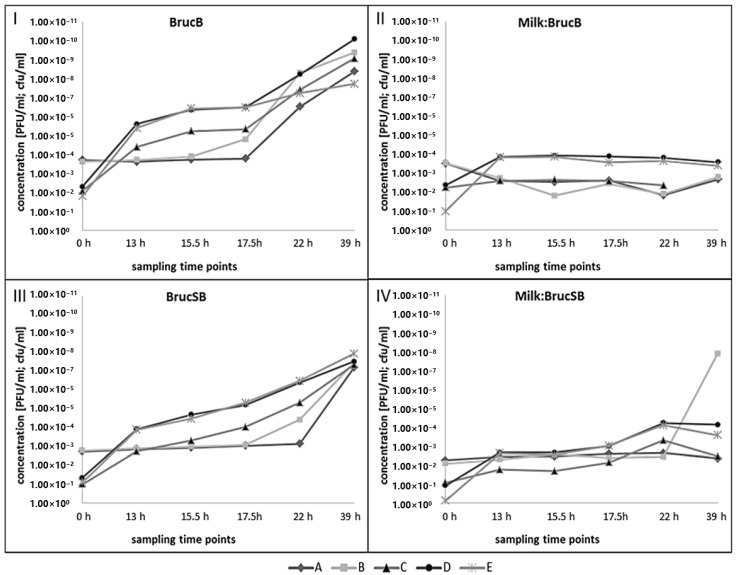
Iz_v_ phage propagation and *B. microti* 4910^T^ growth at 37 °C for 39 h in I—Brucella-broth (BrucB), II—raw cow’s milk in Brucella-broth (1:10, MBrucB), III—selective Brucella-broth (BrucSB), IV—raw cow’s milk in selective Brucella-broth (1:10, MBrucSB) including Iz_v_ phage and *B. microti* in initial concentrations of A + C- 10^6^ PFU/mL and 10^1^ cfu/mL, B + D- 10^6^ PFU/mL and 10^2^ cfu/mL, E- 0 PFU/mL and 10^2^ cfu/mL. A + B show the concentrations of Iz_v_ phage in PFU/mL determined by BPHeli qPCR, C + D + E show the concentrations of *B. microti* in cfu/mL determined by the IS711 qPCR.

**Table 1 microorganisms-08-01326-t001:** Performance data of the BPHeli qPCR (analytical sensitivity).

Iz_v_ Genome Equivalents [log10]	0	1	2	3	4	5	6
Mean Cq-value	33.29	31.06	28.06	24.44	21.01	17.49	13.81
Standard deviation	0.45	0.32	0.21	0.17	0.19	0.17	0.18

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
