# Peer review of "A Proof of Principle for the Detection of Viable Brucella spp. in Raw Milk by qPCR Targeting Bacteriophages"

_microorganisms, 2020, doi:10.3390/microorganisms8091326_

Round 1
Reviewer 1 Report
General considerations:
The article say to develop a highly sensitive and robust qPCR assay for the detection of Brucella phages both in lysates and in raw milk and products thereof but only one phage is really evaluated (Iz) in raw milk and the authors only supposed similar results in other Brucella phages, milk products or conditions without any additional experiment.
The article is well defined and described, and the laboratory experiments supported the showed results but additional experiments should be done with other phages, in other dairy products and in different field conditions, to reach the widespread use of the proposed test.
All experiments are performed in the article with Brucella microti (to be done in laboratories under level 2 biosecurity conditions). On the one hand, B. microti is known to grow faster than classic Brucella species, which cause disease in domestic animals (the authors mention the difference in growth of B. microti on lines 285-286 on page 9) . On the other hand, B. microti naturally infects mice, not domestic mammals, and is not normally found in milk from domestic mammals. Therefore, if the authors want to extrapolate the results to the classic Brucella, I consider it essential to carry out additional experiments on milk and milk products with Brucella species that infect the mammary gland of domestic animals.
What is the advantage of doing the BPHeli qPCR test in tank milk compared to a Brucella qPCR test? How long do Brucella phages remain detectable or alive in milk and milk products even though no Brucella remains alive? Are Brucella phages detected in milk or dairy products from Brucella-free farms? All experiments are performed on the article with a single phage (Iz), and the results are extrapolated to all Brucella phages. In milk and milk products, do all Brucella phages behave the same?
Additionally, today, the typing of isolates with brucelar phages is very low and is only used for research since molecular techniques, such as simple multiple PCR such as the Bruceladder, are much more practical to perform in diagnostic laboratories. What is the advantage of doing the BPHeli qPCR test in Brucella Typing then?
In my opinion, it would only be useful to use the BPHeli qPCR test when performing treatment with Brucella phages on animals infected with Brucella. The release of Brucella phages into Brucella infected milk tanks could have unknown milk quality consequences. And the release of bacteriophages into the environment must always be controlled.
Specific considerations:
In the introduction the classic microbiological methods used for Brucella typing are cited with references 9 and 10. There are other rather better references such as the Bergey´s Manual, OIE Manual, Alton, G. G., L. M. Jones, et al. (1988). "Techniques for the brucellosis laboratory."
In points 2.1 and 2.2 "8,000 rpm" are mentioned. For experiments to be reproducible in other laboratories, the centrifugation speed should be expressed in "g" values.
The experiments have only been done with cow's milk, and should also be extended to sheep's and goat's milk.
Stability experiments of BPHeli qPCR have only been carried out in cultures for up to 39 hours (line 125 page 4). These stability studies should extend over time until the Brucellas were all dead.
The authors point out nonspecific amplifications in three phages (lines 172-173 page 5), which should be mentioned.
The authors say "After 13h we observer a decline of the phage concentration in MBrucB of about 1 log 10 value (lines 197-198 page 7). Is this decrease statistically different?
Reviewer 2 Report
The article of Michaela Projahn et al. concerning the detection of viable Brucella spp. in raw milk by qPCR targeting bacteriophages is of great interest. The authors developed a novel real-time PCR assay targeting the highly Brucella-specific lytic phages. In general, the work is very interesting, and I would like to thank the authors for adding this kind of information to our knowledge. However, there are some minor enquires required to be addressed before publication.
Specific comments
- Page 1, line 29: I suggest to add B. melitensis in the first place to be “are mainly caused by B. melitensis, B. abortus, and B. suis.
- Is the technique efficient with mixed infection of Brucella and α-proteobacterial near neighbors that can be misidentified as Brucella spp., such as Ochrobactrum anthropi and Afipia felis.
- The limit of detection was determined to be lower than 1 genomic copy/μl. In regards to the CFU, how many bacteria can this assay detect in the sample? please translater the knowledge to clarify the lower detectable number of bacteria in the sample.
- In the discussion, lines 277-278, if the arrangement has nothing to tell, please replace B. melitensis which has seven IS711 elements before B. abortus which has only four elements.
Round 2
Reviewer 1 Report
The author should make some changes before the manuscript is accepted
Lines 2-3, title change:
“A prof of principle for detection of Viable Brucella spp in Raw Milk by qPCR Targeting Bacteriophages”.
Lines16 changes:
In artificially contaminated raw milk, in spite of Iz phage was not able to eradicate effectively B. microtti from milk, phages were reliably detected after 39 h of incubation, indicating the presence of viable bacteria.
Lines 17-19 changes:
The qPCR assay showed high stability in the milk matrix and significantly shortened the time to B. microtti diagnosis.
Lines 19-21 changes:
“Hence, our molecular assay is a reliable and sensitive laboratory method to analyze phage titers, but our study reveals infection risks after the phage burst because bacterial cells were still viable implying that Brucella might not be fully eradicated from a food sample using phages. More additional experiments are required to determine the test specificity in field conditions.
Lines 307-312 changes:
We developed a highly sensitive and robust qPCR assay for the detection of Brucella phages both in lysates and in raw milk and products thereof. In this way, our assay indirectly proves the presence of viable Brucella spp. after adding Brucella phages to a contaminated food sample, which is a much faster approach than traditional culture methods. However, after the phage burst bacterial cells were still viable, so Brucella might not be fully eradicated from a food sample using phages. Rapid molecular diagnostics, which are capable of detecting live bacteria, allow for better risk assessment and will significantly improve food safety standards in the future but more additional experiments should be done to determine the test specificity in field conditions.
Round 3
Reviewer 1 Report
I wanted to thank the authors for the changes made to the manuscript, but even so, I believe that the new test proposed in the article is not properly evaluated with field samples, and its specificity is unknown if it is applied in Brucellosis-free herds.
Therefore, I consider it very important that in the "Conclusions" section on line 313 the sentence "More additional experiments should be done to determine the test specificity in field conditions" must be added unless in the “Abstract” section (on line 22) this text is added.